# Melanoma Skin Cancer: A Comprehensive Review of Current Knowledge

**DOI:** 10.3390/cancers17172920

**Published:** 2025-09-05

**Authors:** Camila Caraviello, Gianluca Nazzaro, Gianluca Tavoletti, Francesca Boggio, Nerina Denaro, Giulia Murgia, Emanuela Passoni, Valentina Benzecry Mancin, Angelo Valerio Marzano

**Affiliations:** 1Department of Physiopathology and Transplantation, University of Milan, 20122 Milan, Italy; gianluca.tavoletti@unimi.it (G.T.); angelo.marzano@unimi.it (A.V.M.); 2SC Dermatologia, Fondazione IRCCS Ca’ Granda Ospedale Maggiore Policlinico, 20122 Milan, Italy; giulia.murgia@unimi.it (G.M.); emanuela.passoni@policlinico.mi.it (E.P.); valentina.benzecry@policlinico.mi.it (V.B.M.); 3Pathology Unit, Fondazione IRCCS Ca’ Granda Ospedale Maggiore Policlinico, 20122 Milan, Italy; francesca.boggio@policlinico.mi.it; 4SC Oncologia Medica, Fondazione IRCCS Ca’ Granda Ospedale Maggiore Policlinico, 20122 Milan, Italy; nerina.denaro@policlinico.mi.it

**Keywords:** melanoma, epidemiology, histopathology, staging, risk factors, systemic therapy, clinical trials

## Abstract

Cutaneous melanoma is an aggressive form of skin cancer, accounting for the majority of skin cancer-related deaths. Its global incidence has been increasing over the last few decades. Knowledge about cutaneous melanoma has advanced considerably. However, updated information about the disease is often scattered across various sources. Given the importance of this complex disease, in this review, we aim to provide updated and comprehensive knowledge on melanoma in a single, accessible source. This paper covers key aspects of melanoma skin cancer—from molecular biology, histopathology, and genetics to epidemiology, diagnosis, staging, and treatment. It also highlights emerging technologies and future directions in the management of melanoma skin cancer.

## 1. Introduction

Cutaneous melanoma is a skin cancer caused by the abnormal proliferation of melanocytes in the epidermis. Its incidence has been increasing during the last decades [1,2], mainly due to increased ultraviolet (UV) radiation exposure [3]. The disease is most commonly suspected on a changing pigmented lesion and confirmed with histopathological analysis.

Melanoma management has advanced significantly over the past decade, with considerable impact on the prognosis of the disease. Improved surveillance, widespread use of sentinel lymph node (SLN) biopsy, and advances in systemic therapy have all contributed considerably to the decrease in melanoma mortality [4].

This review provides a comprehensive discussion on cutaneous melanoma, aiming to build a solid foundation for understanding the disease and supporting clinical decision-making. From essential aspects of histopathology and genetics to insights into epidemiology and patient management, it examines current evidence and explores future directions on this complex disease.

## 2. From Melanocytes to Melanoma: Molecular Biology and Genetics

Melanocytes are pigment-containing cells derived from neural crest stem cells, which produce melanin, determining hair and skin color. In the skin, melanocytes are stimulated by different factors, including UV radiation, to convert tyrosine into melanin [5,6].

Melanocyte malignant transformation is a result of stepwise accumulation of genetic mutations initiated by gain-of-function mutations in specific oncogenes (e.g., *BRAF*, *NRAS*, *GNAQ*, *GNA11*, *KIT*) [7,8,9,10,11,12,13]. At this point, transformation can still be discontinued by senescence mechanisms. Secondary loss-of-function mutations in tumor suppressor genes (e.g., *CDNK2A*, *TP53*, *PTEN*, *BAP1*, *NF1*) [7,14,15,16] overcome those obstacles and establish the progression from benign lesions to intermediate or malignant tumors [7]. The above mutations can be inherited or somatically acquired, and some are found in predisposing syndromes to diverse tumors, including pancreatic cancer, mesothelioma, and meningioma [7,13,17,18].

Mutations related to melanoma most frequently activate the mitogen-activated protein kinase (MAPK) pathway, a cell proliferation pathway associated with oncogenic signaling. Another relevant pathway is the phosphatidyl inositol 3-kinase (PI3K), responsible for cell survival. Mutations in the *BRAF* gene are the most frequent somatic mutations in melanoma, activating specifically the MAPK pathway. The second most common mutations are *NRAS* mutations, which activate both the MAPK and PI3K pathways [12,19].

## 3. Pathophysiology

After melanocyte transformation, melanoma spreads in two different phases: radial and vertical growth.

Melanoma usually arises as a superficial tumor in the epidermis and initially grows horizontally, in what is called the radial growth phase (RGP). In this phase, melanoma is mostly curable by surgical excision alone. RGP melanomas can be further divided into “Melanoma in situ” and “Microinvasive radial growth phase melanoma”. “Melanoma in situ” is characterized by the proliferation of melanocytes in the epidermis only, whereas “Microinvasive radial growth phase melanoma” involves microscopic dermal invasion. At this degree of dermal extension, the largest dermal nest cannot be larger than epidermal nests, and there should be no presence of dermal mitosis [20,21,22,23,24].

After acquiring genetic mutations and molecular changes, melanoma enters the vertical growth phase (VGP) and infiltrates deep into the dermis. At this point, the tumor has metastatic potential. By definition, in the VGP, the tumor has at least one dermal nest larger than the largest epidermal nest or the presence of dermal mitosis. The invasion of the dermis can occur later, also through expansile nodules, and the tumor may further expand to the reticular dermis or fat tissue [20,21,23,24,25].

The depth of tumor invasion can be described by the Clark [26] and Breslow classification systems [27]. Clark levels are briefly defined as Level I (tumor above the basement membrane—melanoma in situ); Level II (tumor extends into the papillary dermis); Level III (melanoma extends to the papillary–reticular dermis interface); Level IV (distinct invasion into the reticular dermis); and Level V (invasion of the subcutaneous tissue) [26]. In the VGP, the invasion of reticular dermis or fat tissue corresponds to Clark levels III, IV, and V, respectively.

Breslow staging [27], on the other hand, is a measure, in millimeters, of the depth of invasion of the nodule beneath the top of the granular cell layer of the epidermis. It is highly reproductible and better correlates with patient prognosis compared to the Clark level of invasion [28,29]. Breslow staging is used to determine the tumor (T) stage in the American Joint Committee on Cancer (AJCC) staging system [30].

## 4. Subtypes: Morphology and Histopathology

The different melanoma subtypes are distinguished by histology and morphological characteristics of the growth phase. Classically, Clark [24] divided melanoma into four main subtypes: superficial spreading melanoma (70% of all melanomas), nodular melanoma (approximately 15%), lentigo maligna melanoma (approximately 10%), and acral lentiginous melanoma (approximately 2% of all melanomas) [24,31,32,33,34].

### 4.1. Superficial Spreading Melanoma

Superficial spreading melanoma (SSM) is the most common subtype of melanoma. The surface of the tumor is either a macule or plaque with an irregular border, which ranges in size up to centimeters. The color of the lesion can include shades of red (representing vessel ectasia and inflammation), blue, black, gray, and white (amelanotic or regressed foci) [26,31,35] (Figure 1a,b).

Histologically, SSM is characterized by asymmetric, poorly demarcated melanocytes that lack cellular maturation. Melanocytes are distributed unevenly, often with solitary melanocytes being more predominant over nests, and commonly with evidence of intraepithelial spread in a pagetoid manner [35,36,37] (Figure 2).

Most superficial spreading melanomas are de novo lesions; however, 30% of cases occur in association with a pre-existing nevus, such as a dysplastic or congenital nevus [38].

### 4.2. Lentigo Maligna Melanoma

Lentigo maligna melanoma (LMM) usually arises in sun-damaged skin as a result of cumulative chronic exposure to UV radiation [32,39]. In the early stages, it appears as a tan-brown macule, similar to a freckle. It slowly increases in size and darker, asymmetric foci appear (Figure 3a,b). After several years to decades, a small percentage of lentigo maligna (approximately 5%) slowly invades into the dermis, transforming itself from melanoma in situ lentigo maligna type to lentigo maligna melanoma. Clinically, this transformation can be seen as a lesion that becomes palpable [26,40,41,42].

Histologically, lentigo maligna melanocytes are usually hyperchromatic and array along the dermal epidermal junction in a lentiginous pattern. Frequently, neoplastic proliferation extends to the follicular epithelium with moderate atypia and pleomorphism. The epidermis is usually atrophic, and the dermis displays signs of solar elastosis. When dermal infiltration occurs, neoplastic melanocytes often assume a spindle-shape morphology [26,32,37,40,41] (Figure 4).

### 4.3. Nodular Melanoma

Nodular melanoma is defined as melanocyte proliferation with vertical growth, which can occur in association or not with an epidermal component. Therefore, an “in situ” form of nodular melanoma cannot exist, as the proliferation of this variant is never restricted to the epidermis.

Clinically, nodular melanoma usually appears as a brown/gray/black pedunculated or polypoid nodule [26,37] (Figure 5a,b).

Histologically, if an intraepidermal component is present, it should not extend for more than three rete ridges beyond the dermal component (otherwise, it is considered the vertical growth phase of a superficial spreading melanoma) [26]. These lesions frequently show dermal mitosis and epidermal ulceration (Figure 6).

### 4.4. Acral Lentiginous Melanoma

Acral lentiginous melanoma originates on the extremities of the body (palms, soles, and nail units). However, not all melanomas in acral sites are acral lentiginous melanoma; approximately 50% correspond to other subtypes [43]. It is predominant among patients with non-white skin tones [33].

Clinically, acral lentiginous melanoma presents as a dark brown or black pigmented lesion (Figure 7a,b). In the nail unit, it usually appears as a longitudinal melanonychia, with or without nail dystrophy (Figure 8). Hutchinson’s sign, the involvement of the nail fold, is an important feature for diagnosis (Figure 8). The raising of a lesion or development of ulceration typically indicates invasion of the dermis. However, even flat acral lentiginous melanoma can already have invaded the adventitial dermis [41,44,45].

Histologically, melanocytes are usually hyperchromatic. Initially, melanocyte cells are arranged in a lentiginous pattern at the basal epidermal layer. With progression, lesions frequently show more frequent nests, intraepithelial pagetoid spreading, and spindle cell morphology [24,37,41] (Figure 9).

Besides these four main subtypes of melanoma, other variants exist (e.g., nevoid, desmoplastic, spitzoid melanoma) and represent a challenge for correct diagnosis, both clinically and histologically, as they can mimic benign lesions [7,46,47].

### 4.5. The 2022 World Health Organization Classification—Fifth Ed

The fifth edition of the World Health Organization (WHO) Classification of Skin Tumours [48], updated in 2022, reaffirmed the multidimensional pathway classification proposed by Bastian et al., introduced in the fourth edition [18,49]. Based on epidemiologic, clinical, histopathologic, and genomic features, melanocytic tumors are outlined into nine evolutionary pathways (under three different categories based on the intensity of chronic ultraviolet radiation exposure/cumulative solar damage (CSD)) (Table 1).

The term “melanocytoma” remains an intermediate category of tumors between nevi and melanoma. The term is applied to neoplasms with more than one driver mutation, with a second mutation affecting specific pathways that result in distinct, atypical microscopic features and increased risk of local recurrence. Examples of melanocytoma include pigmented epithelioid melanocytoma (PEM), WNT-activated deep penetrating/plexiform melanocytoma (previously classified as deep penetrating nevus), and *BAP1*-inactivated melanocytoma [48].

A summary of the WHO classification of melanocytic tumors can be found in Table 1.

## 5. Epidemiology

Melanoma is the most severe form of skin cancer, responsible for over 80% of skin cancer mortality [1,50]. Fortunately, it accounts for only 1 percent of skin cancer [1]. Worldwide, it is responsible for approximately 325,000 new cases and 57,000 deaths annually [3].

The incidence of cutaneous melanoma has been increasing during the last decades [1,2] and is projected to more than double in the next 20 years [3]. This is mainly associated with increased UV radiation exposure [3]. Despite the overall increase in incidence, melanoma’s incidence in young people has been decreasing in the United States and Australia, pointing to the benefits of sun-protective behavior prevention programs [51,52].

UV radiation exposure is considered the main risk factor for melanoma’s development (around 75% of all melanoma cases are linked to exposure to UV light [53]). Melanoma is primarily affected by intermittent intense sun exposure, especially sunburns during childhood and adolescent years [54,55,56,57,58]. UV exposure through indoor tanning also represents an important risk factor for the development of the disease [57,59].

The risk of developing melanoma increases with skin color [60] (Fitzpatrick Skin Phototype Classification 1–3 [61]), age [3] (>50 years old, especially older than 80 years old), sex [62] (females under 40 years old and in males over 65 years), immunosuppression [63,64], personal history of nonmelanoma skin cancer [65], and positive family [60] or personal history of melanoma [60,62,66,67,68].

Additional risk factors for cutaneous melanoma include an increased number of nevi, the presence of atypical nevi, large congenital melanocytic nevi, and atypical mole syndrome (AMS)/familial atypical multiple mole and melanoma (FAMMM) syndrome [69,70,71]. AMS is a spectrum of phenotypic expressions characterized by the presence of multiple moles (often >50), some of which are atypical (dysplastic nevi). This condition can be sporadic or hereditary [72]. When hereditary with one or more first- or second-degree relatives affected, it is called FAMMM syndrome [73]. FAMMM syndrome is an autosomal dominant condition highly associated with mutations in the *CDKN2A* gene [74,75].

Occupational exposure to chemicals and ionizing radiation [76] and personal history of specific diseases, such as Parkinson’s disease [77,78,79] and endometriosis [80], have also been associated with increased melanoma risk in some studies.

Earlier melanoma detection is relevant in order to improve prognosis and provide a more cost-effective treatment for patients and healthcare institutions. A routine visual screening exam is usually the screening strategy of choice and is recommended for asymptomatic individuals with high risk for skin cancer [62]. Because of the absence of a randomized melanoma screening trial and lack of consistent evidence of a decrease in mortality due to melanoma population-based screening programs in observational studies, there is insufficient data to recommend melanoma screening for low-risk individuals [62,74,75]. There is still no consensus on the optimal melanoma screening strategy for the general population. Most healthcare systems, including those from the United Kingdom, Australia, and Italy, do not recommend population-based routine melanoma screening [74,75]. In Germany, on the other hand, since the results of the SCREEN project [81], patients insured by statutory health insurance (the majority of the population) are eligible for biennial full-body skin cancer screening examination [82,83]. Following the introduction of this nationwide screening program, however, there has not been a significant decline in melanoma-related mortality in the country [84].

## 6. Diagnosis

### 6.1. Clinical Diagnosis

A clinical concerning lesion is usually a changing/growing pigmented lesion (although it can also be amelanotic). Different criteria have been suggested in order to detect suspicious lesions. The ABCDE criteria [85] (Figure 10), based on the evolution of the radial growth phase, have been a fundamental resource in aiding melanoma earlier detection. Sensitivity of the individual ABCDE criteria is described as 57, 57, 65, 90, and 84% and specificity as 72, 71, 59, 63, and 90%, respectively [86]. Challenges of the ABCDE criteria include the recognition of small lesions (<6 mm) [87], as well as the recognition of specific melanoma subtypes that commonly lack ABCD features, such as nodular, amelanotic, and desmoplastic melanoma [88,89,90].

To overcome those limitations, other criteria have been proposed for clinical evaluation, including the ugly duckling sign (presence of a lesion that does not match the patient’s nevus phenotype) [91], the revised seven-point checklist [92,93] (Table 2), and the overall irregular pattern of a lesion [49,94,95].

To improve diagnostic accuracy, dermoscopy is also used by trained physicians to individualize melanoma from nonmelanoma suspicious skin lesions [96]. Melanoma dermoscopic characteristics include multiple colors asymmetrically distributed, an atypical pigment network, atypical globules, pseudopods, a negative network, a blue-white veil, an atypical vascular pattern, and shiny white structures [31,97,98,99].

Non-invasive imaging techniques have also been proposed, in conjunction with clinical and dermoscopic examination, to aid in differentiating malignant from benign melanocytic lesions [100]. Newer technologies include reflectance confocal microscopy [101], optical coherence tomography [102,103,104], multiphoton microscopy [104], and dermatofluoroscopy [105]. Operator dependence, cost-effectiveness, and accessibility still limit the implementation of these diagnostic tools in clinical practice.

### 6.2. Diagnosis Confirmation

An excision biopsy is recommended for all suspicious lesions, as histopathology is the gold standard for melanoma diagnosis. Skin biopsy should be performed with 1–3 mm margins that encompass all the lesion with clinically negative margins, until a depth sufficient to ensure that the lesion is not histologically transected at the deep margin. When the complete lesion cannot be excised (e.g., large lesions, uncertain diagnosis with low melanoma clinical suspicious, or challenging anatomic sites, including face and acral sites), partial/incisional biopsy may be acceptable. In those circumstances, the biopsy should include the most atypical portion of the lesion [106,107].

To assist pathologists when interpreting the specimen, clinicians should consider patient’s sex and age, size and site of the lesion, duration and changes in the lesion over time, dermoscopic features, previous trauma or prior biopsy, pregnancy status, clinical impression and differential diagnosis, type of biopsy performed, and, if available, clinical photographs of the lesion [106,108].

Pathological features are of extreme importance to confirm the diagnosis and guide management of the disease. Besides the histologic subtype, it is important to evaluate other prognostic factors. These include Breslow thickness, presence of ulceration, margin status, and, for lesions in the vertical growth phase, the presence of lymphocytic infiltrates, microsatellites, mitoses, and neural or vascular invasion [13,107,108].

In challenging cases, it may be useful to add immunohistochemistry and molecular tests to the histopathological diagnosis, including comparative genomic hybridization, fluorescence in situ hybridization (FISH), targeted gene sequencing, and gene expression profiling [13,107].

## 7. Staging

Staging should be performed in all patients with a new diagnosis of melanoma. The most recent American Joint Committee on Cancer (AJCC) staging system is the eighth edition [30].

Staging is divided into clinical and pathological. Clinical staging is based upon the initial primary tumor biopsy and clinical/radiologic assessment of regional lymph nodes and distant metastases. Pathological staging includes the worst features of the primary tumor biopsy and surgical excision specimens, as well as pathological information from SLN biopsy or therapeutic lymph node dissection. Stage 0 corresponds to melanoma in situ. Stages I and II indicate localized invasive disease, while stage III represents regional metastasis and stage IV, distant metastatic disease [30].

### 7.1. T Category: Primary Tumor

The T category is divided into T1–T4 based on the tumor thickness measured in mm (Breslow staging) and should be recorded in the pathology report to the nearest 0.1 mm, rather than 0.01 mm (a change from the AJCC’s previous edition) [109]. Tumor thickness is measured from the top of the granular layer of the epidermis or from the base of the ulcer up to the deepest invasive cell in the dermis or subcutis [30,108]. T0 designates an unknown or completely regressed primary tumor. Tis corresponds to melanoma in situ, and TX is used when tumor thickness cannot be determined.

Each T category is also subdivided into a and b, based on the presence of ulceration, defined as the absence of an intact epithelium over the melanoma. Ulceration represents a negative prognostic factor in patients with the same tumor thickness.

Unlike the seventh edition [109], the mitotic rate is no longer described as a dichotomous subcategory separating T1a and T1b. Nevertheless, it is still, across its dynamic range, an independent predictor of adverse outcome in melanoma patients. The mitotic rate is assessed by the “hot spot” method and should continue to be recorded in pathology reports because of its prognostic value.

### 7.2. N Category: Nodal Involvement

In the AJCC staging system eighth edition, regional lymph node involvement is characterized as being clinically occult (positive sentinel lymph node biopsy microscopic findings) or clinically detected (positive physical examination or imaging) [30,108]. These terms differ from the previous “microscopic” and “macroscopic” terms used to describe regional node metastases in the seventh edition [109].

The N category indicates metastasis-containing regional lymph nodes. Patients can also be assigned an N “c” subcategory if non-nodal regional metastases are present, including microsatellites, satellites, or in-transit metastases [4].

Sentinel lymph node biopsy is required for pathological staging. All patients with histologic confirmation of lymph node involvement without distant metastasis are classified as stage III disease.

### 7.3. M Category: Distant Metastasis

Patients with distant metastasis are subcategorized into M1a, b, c, or d based on the site of disease involvement. A difference from the AJCC staging system, seventh edition, includes the suffixes 0 or 1 for each M subcategory to indicate elevated (1) or not elevated (0) serum lactate dehydrogenase (LDH) [30,108]. Serum LDH was demonstrated to be an independent prognostic factor in patients with disseminated melanoma.

## 8. Prognosis

Early diagnosis of melanoma is of extreme importance to reduce mortality, as treatment at an early clinical stage is associated with a great prognosis. Melanoma’s 5- and 10-year survival rates range from 99% and 95% in stage 1 to 90% and 84% in stage 2 and 77% and 69% in stage 3, respectively [4].

The survival rate has increased considerably during the last decade [110,111]. One of the reasons is the widespread use of SLN biopsy, required for staging patients with T2 through T4 primary melanoma, which was responsible for correctly staging patients with clinically occult nodal metastasis who previously might have been misclassified as node negative [4]. Advances in systemic therapy with immune checkpoint inhibitors and *BRAF*-targeted therapies were also responsible for increasing survival rates in advanced melanoma [110,111]. At the beginning of the year 2000, patients with stage IV melanoma had around a 50% 1-year survival rate [112]. Twenty years later, the 1-year survival rate is more than 70% for stage IV patients treated with targeted and immunotherapies [108,113].

### Prognostic Factors

Clinical and pathological characteristics are associated with different melanoma outcomes.

Clinically, older age is related to thicker primary tumors, a higher mitotic rate, and ulceration. Patients older than 70 years old have the most aggressive tumor findings and reduced survival [114]. Male sex is another independent risk factor for worse overall survival [115]. Moreover, primary melanomas located in the head/neck and trunk are also related to worse prognosis [116,117].

Histopathologic factors also significantly impact survival. The presence of ulceration is correlated with a worse melanoma outcome. According to the database analysis that supported the AJCC eighth edition, an ulcerated primary melanoma was associated with survival rates similar to those of a patient with a nonulcerated primary in the next highest T category. The mitotic rate, when evaluated as its dynamic range, was also considered a relevant prognostic determinant associated with increased risk of SLN metastasis. Regarding nodal involvement, increasing SNL tumor burden is related to reduced survival [4].

Molecular techniques have been proposed as predictive tools for melanoma prognosis and patient outcomes. Circulating melanoma cells and circulating tumor DNA (ctDNA) in the peripheral blood have been related to an increased risk of recurrence and worse survival [118,119,120]. Gene expression profiling (GEP) and proteomics have also been pointed out by some studies to improve risk prediction [121,122,123]. However, these techniques are still recent, and further data are necessary to support their routine use in clinical practice [13,124,125].

Not all relevant prognostic factors have been included in the American Joint Committee on Cancer staging system. Data are continuously evolving, and prognostic algorithms are being developed to enhance the prediction of melanoma-specific survival, the presence of regional and distant metastasis, and melanoma site-specific recurrence [4].

Another current limitation of the AJCC staging system is that it does not consider conditional survival estimates. Prognostic estimates are defined at the time of diagnosis and are not updated over time. Adding estimates of conditional survival to the staging system may provide prognostic information that helps to individualize clinical practice [126].

## 9. Management

Following the diagnosis of melanoma, a comprehensive patient history with a complete review of systems and physical examination should be performed.

Physical examination includes total body skin examination with evaluation of the biopsy site and surrounding skin for local or satellite/in-transit metastases and evaluation of regional and distant lymph node basins [106,127]. The findings of a physical examination direct the need for further laboratory and imaging studies.

Patients should also be instructed to perform regular skin self-examinations to detect any recurrence site, satellite/in-transit metastases, new primary lesions, or enlarged lymph nodes [106,127]. Additionally, patients should be educated in the principles of sun safety to avoid new primary melanomas [107].

### 9.1. Imaging and Laboratory Studies

For baseline evaluation of newly diagnosed asymptomatic patients with melanoma, laboratory and imaging studies’ indications vary based on the staging of the disease. For each category, it is important to take into consideration prognosis and risk for metastatic disease, as well as imaging studies’ high false-positive rate, costs, and patient anxiety [106,107,127].

Patients with high-risk melanoma (stages IIB and IIC) and advanced melanoma should be evaluated by a multidisciplinary team in collaboration with medical oncology [106].

For asymptomatic patients with melanoma in situ, stage IA, no routine laboratory or imaging studies are recommended [107,127].

For asymptomatic patients with stage IB and higher, the latest European consensus-based interdisciplinary guideline recommends the performance of ultrasound of locoregional lymph nodes [127]. The American Academy of Dermatology and the NCCN Guidelines recommend requesting an ultrasound if unable to assess or altered nodal involvement on physical examination, unless needed for surgical planning or initiation of systemic treatment [106,107]. Abnormal or suspicious findings should be confirmed with core biopsy (preferred) or fine-needle aspiration.

For stages IIB and IIC patients who are candidates for adjuvant treatment, it is also recommended to obtain systemic imaging. Systemic imaging includes a CT scan of the chest and abdomen or whole-body positron emission tomography (PET-CT) [107]. Some guidelines may also recommend magnetic resonance imaging (MRI) of the brain to exclude central nervous system (CNS) involvement [127].

Patients with clinically detected nodal disease or in-transit/satellite metastases are classified as stage III disease. For those patients, it is recommended to histologically confirm nodal involvement with core biopsy (preferred) or fine-needle aspiration and obtain systemic imaging with computed tomography (CT scans) or whole-body PET-CT, together with magnetic resonance imaging of the brain [107,127].

For patients with symptoms, clinically suspected systemic metastases, or stage IV disease, it is recommended to obtain total body imaging using CT or PET-CT and brain MRI [107,127]. For patients with stage IV disease, it is also recommended to obtain LDH levels for correct staging [30].

### 9.2. Surgery

Management of cutaneous melanoma involves wide excision of the primary melanoma site with a margin of normal tissue. Definitive excision should be performed within 4 to 6 weeks of initial diagnosis [128].

Surgical margins are radial and measured clinically by the surgeon from the edge of the biopsy site or residual intact component. The lesion should be excised to the depth of, without including, the muscular fascia. The margin is determined based on the tumor thickness [106,128,129,130]. In constrained areas, such as the head and neck or acral sites, margins might not always be attainable. For minimally invasive (≤T1a) melanoma in those areas, especially melanoma in situ lentigo maligna type, another surgical possibility is Mohs micrographic surgery (MMS) [106,107,128,131].

MMS is a tissue-sparing technique developed by Dr. Frederic Mohs [132,133]. The procedure involves excising a thin layer of tissue circumferentially from around and beneath the clinical margins of the tumor. The specimen is then processed with frozen, horizontal sections, allowing for immediate histological evaluation by the Mohs surgeon. The process is repeated at the positive margins until negative histologic margins are confirmed. This technique allows for precise microscopic examination of the entire tumor margin while preserving the maximum amount of healthy tissue [134]. MMS is not considered standard care for invasive melanoma (T1–T4) [106,107,131].

### 9.3. Non-Surgical Approaches

For patients with melanoma in situ lentigo maligna, when a surgical procedure is not possible or will lead to relevant disfigurement, a second-line treatment includes primary radiotherapy or topical imiquimod 5% [106,128,135,136]. For those patients, topical imiquimod can also be used as adjuvant therapy after surgery [106,128,136,137].

For desmoplastic melanoma with high-risk features, radiotherapy can be used as an adjuvant treatment based on local recurrence risk [106,107].

There is limited evidence for second-line treatments. Therefore, patients should be carefully selected, and the choice of treatment should be discussed between the patient and the physician.

### 9.4. Lymphatic Mapping and Sentinel Lymph Node Biopsy

Lymphatic mapping and SLNB are recommended for patients with clinically negative lymph nodes based on the risk of nodal metastases.

Tumor thickness is the primary feature to predict nodal involvement risk. Lymphatic mapping and SLNB are always recommended for patients with Breslow thickness > 1 mm (T2–T4b, >10% risk of regional metastasis). For patients with a 5–10% risk of regional metastasis (melanoma thickness 0.8 to 1 mm, <0.8 mm with ulceration or 0.5–0.8 with high-risk features), SLNB should be considered [128]. High-risk features include a high mitotic rate, younger patients, lymphovascular invasion, and head/neck locations [107,128]. If the thickness is lower than 0.8 mm and the lesion is nonulcerated (T1a with no adverse features, <5% risk of regional metastasis), SLN biopsy is usually not recommended [107].

When indicated, sentinel lymph node biopsy should be performed before wide excision of the primary tumor and in the same operative setting, whenever possible.

### 9.5. Additional Treatments

#### 9.5.1. Negative Sentinel Lymph Node Biopsy

For patients with a negative sentinel lymph node biopsy, surgical excision alone is curative in most cases. However, there is a group of patients with a higher risk of recurrence that benefits from systemic therapy.

For patients with prognostic stages I or IIA and a negative SLNB, treatment includes surgical excision only without the need for adjuvant therapy [106,107,128].

For patients at higher risk of recurrence and metastatic disease (stages IIB and IIC), one year of adjuvant immunotherapy treatment with programmed cell death receptor 1 (PD-1) blockade (pembrolizumab or nivolumab) is recommended. In recent clinical trials, both treatments have been demonstrated to reduce the risk of recurrence and improve distant metastasis-free survival [138,139,140].

#### 9.5.2. Positive Sentinel Lymph Node Biopsy

For patients with a positive sentinel lymph node biopsy, clinical observation of the lymph node basin with ultrasound follow-up is the standard care. As demonstrated by the DeCOG-SLT and MSLT II trials [141,142], the treatment of individuals with a positive SLNB with complete lymph node resection (CLND) is related to increased morbidity without a benefit in survival. CLND should be reserved for specific patients with a higher SLN tumor burden and adverse histological characteristics of the primary lesion that could predict a higher probability of additional positive lymph nodes [107,128].

Adjuvant systemic therapy is also usually indicated, based on the higher risk of recurrence of stage III patients. For stage IIIA patients with very small SLN tumor deposits only, systemic therapy could be avoided because of the low risk of disease recurrence [107,128,143,144]. For all other stage III patients, one year of adjuvant systemic therapy is indicated. Therapy is based upon the *BRAF V600* mutation status. *BRAF* wild-type tumors should be treated with PD-1 inhibitors (nivolumab or pembrolizumab) [145,146], whereas *BRAF*-mutant tumors can be treated with either a PD-1 inhibitor or targeted therapy with BRAF plus MEK inhibitors (dabrafenib plus trametinib) [145,146,147,148].

#### 9.5.3. Clinically Detected Regional Lymph Nodes

Patients with clinically detected regional lymph node involvement, confirmed cytologically or histologically and without distant metastases, should undergo wide excision of the primary lesion and therapeutic lymphadenectomy [107]. Therapeutic lymph node dissection is a procedure with a risk of complications, including dehiscence, infection, and lymphedema [106].

Prior to lymph node dissection, neoadjuvant therapy is considered, followed by surgery and adjuvant treatment. In order to limit therapeutic lymph node dissection surgical morbidity, limited nodal resection has been proposed for selected patients after neoadjuvant immunotherapy [149].

Standard regimens of neoadjuvant therapy include the use of pembrolizumab or nivolumab plus ipilimumab [150,151,152,153]. Smaller studies also demonstrated the efficacy of alternative neoadjuvant therapy with relatlimab plus nivolumab [154]. For *BRAF*-mutant tumors, the use of neoadjuvant targeted therapy with BRAF plus MEK inhibitors can be a possibility. Target therapy, however, was associated with reduced recurrence-free and overall survival when compared to immunotherapy [155].

#### 9.5.4. Distant Metastatic Disease

Patients with stage IV disease should always be discussed by interdisciplinary tumor boards. Several factors should be taken into account for treatment selection, including patient characteristics (age, comorbidity, symptoms, need for rapid tumor size reduction, family/caregiver support, compliance with oral medication) and tumor characteristics (site and number of metastases, *BRAF* status, PD-L1 expression).

For stage IV patients, systemic therapy is indicated. For oligometastatic disease, metastases-directed treatment is an option associated with adjuvant immunotherapy. Preferred regimens with PD-1-based therapy include nivolumab, pembrolizumab, or nivolumab plus ipilimumab [107,128,145,156,157]. For patients with unresectable disease and naive to treatment, initial immunotherapy with a PD-1 agent is also recommended. Different regimes include pembrolizumab [158], nivolumab [159], nivolumab plus ipilimumab [160,161], and nivolumab–relatlimab [162,163]. Triple immune checkpoint therapy with nivolumab + relatlimab + ipilimumab was also investigated in a small cohort of patients (RELATIVITY-048 trial), with an objective response rate of 58.7% and a 48-month overall survival rate of 71.7% [164]. For patients with *BRAF V600* mutation, other recommended regimens include BRAF plus MEK inhibitors [107].

For patients with progressive disease with cutaneous and subcutaneous lesions, locoregional disease, or in-transit metastasis, a promising area of ongoing research is the use of intralesional/intratumoral injections with oncolytic viruses and cytokines [165,166]. The goal of those therapies is to directly kill tumor cells or increase the antitumoral immune response. They can be applied alone or in combination with systemic therapy.

The first and only oncolytic viral therapy to be approved by the United States Food and Drug Administration for local treatment of unresectable, cutaneous, subcutaneous, and nodal lesions in patients with recurrent melanoma is Talimogene laherparepvec (T-VEC). T-VEC is a live attenuated genetically modified herpes simplex virus type 1 (HSV1) that selectively replicates within tumor cells, inducing a tumor-specific immune response. In the OPTiM randomized open-label phase III trial, T-VEC demonstrated significant improvement in overall survival and complete response rate [167]. T-VEC is generally well tolerated and achieves the highest response rates in early metastatic disease [168]. Therapy efficacy seems to increase in combination with immune checkpoint inhibitors. In the last decade, clinical trials have investigated the combination of T-VEC with ipilimumab, pembrolizumab, and nivolumab [169,170,171,172,173]. In a randomized phase II study, where 198 patients were randomized to receive T-VEC plus ipilimumab or ipilimumab monotherapy, combination therapy showed higher objective response rates [169]. Five-year outcomes from the same study confirmed durable response rates [170]. T-VEC has also demonstrated efficacy when reintroduced after disease recurrence in patients who previously achieved an initial complete response with T-VEC monotherapy [174]. Recently, T-VEC has been studied as a neoadjuvant therapy [173,175,176,177].

Similar to T-VEC, vusolimogene oderparepvec (RP1) is a next-generation intratumoral HSV1-based oncolytic immunotherapy. Results from the IGNYTE trial [178] demonstrated durable systemic responses after RP1 therapy combined with nivolumab in patients with advanced melanoma that progressed on anti-PD-1. In addition to T-VEC and RP1, promising oncolytic viral therapies include Canerpaturev (C-REV, formerly HF10), coxsackievirus A21 (CAVATAK), and Lerapolturev (PVSRIPO) [179,180,181].

Besides oncolytic viral therapies, other investigational agents include cytokine intratumoral injections. Daromun, a combination of two immunocytokines, L19IL2 and L19TNF, showed promising results in the PIVOTAL trial for resectable, locally advanced stage III melanoma [182]. Currently, in the INTACT/MeRCI study (NCT06284590), the combination of L19IL2 or L19TNF or L19IL2/L19TNF with pembrolizumab is being investigated for the treatment of stages III and IV unresectable melanoma [183].

Patients with stage IV progressive disease should receive therapies taking into account individual characteristics, including performance status and comorbidities. Those patients should also be encouraged to enroll in clinical trials.

### 9.6. Surveillance

Surveillance follow-up recommendations vary according to the risk of disease recurrence and may have slightly different schedules among international guidelines. Disease recurrence and metastasis usually occur in the first 3–5 years after initial treatment [184]. Moreover, during the same time period, around 4–8% of patients develop a new primary lesion [106,185].

For patients with stage 0 disease, dermatologic follow-up visits are recommended every 6–12 months during the first 1–2 years and yearly after that. During those visits, physical examination should focus on the assessment of local recurrence and the detection of new primary cutaneous melanoma. For stages IA-IIA, it is also recommended to perform a comprehensive history and review of symptoms, including signs or symptoms of disease recurrence. For these patients, physical examination focuses on the evaluation of regional lymph nodes in addition to skin inspection. Visits should be performed every 6–12 months for the first 5–10 years and annually thereafter. For patients with stage IIB or higher, multidisciplinary surveillance is recommended in collaboration with medical oncology, and laboratory and imaging tests are recommended for follow-up. For those patients, follow-up is usually recommended every 3 months for the first 2–3 years, every 6 months up to the 5th–10th year, and annually after that [106,107,127]. In addition to physical examination, imaging studies can be considered for the first 2–5 years. Imaging follow-up includes CT scans of the chest, abdomen, and pelvis or positron emission tomography PET-CT, and regional nodal ultrasound for patients with a positive SLNB without lymph node dissection [186,187].

The European consensus-based interdisciplinary guideline also recommends routine measurement of serum S100β for the detection of recurrent disease in the first few years of follow-up for patients with stages IB-IIA or higher [127,188,189]. Serum S100β measurement is not routinely recommended for surveillance of asymptomatic patients with melanoma in the United States [106,107,188,190].

## 10. Genetic Counseling

Patients with newly diagnosed cutaneous melanoma might require genetic counseling and testing for familial melanoma. Familial melanoma should be suspected in patients diagnosed with melanoma at an early age who have multiple primary melanomas (three or more cutaneous melanomas), have multiple atypical moles or dysplastic nevi, and have a personal history or ≥2 relatives affected by associated tumors (pancreatic, breast, or brain tumors) or a family history of melanoma [106,107].

*CDKN2A* is the most frequent pathogenic variant in familial melanoma and is also associated with pancreatic cancer. In high melanoma incidence areas, genetic counseling and *CDKN2A* testing are recommended for individuals with three or more primary invasive melanomas or families with at least one invasive melanoma and two more cases of melanoma and/or pancreatic cancer among first- or second-degree relatives on the same side of the family [191].

Cutaneous melanoma is also observed at higher rates in other hereditary syndromes associated with mutations in tumor susceptibility, including xeroderma pigmentosum (multiple *XP* genes) and Li Fraumeni syndrome (*TP53* mutations) [192,193]. Additional pathological variants associated with melanoma include *BAP1*, *BRCA2*, *CDK4*, *MITF*, *PTEN*, *POT1*, and *RB1* [194,195,196,197,198,199,200]. Individuals carrying these mutations should receive regular skin screening examinations and be educated on photoprotection and self-skin examination [197].

## 11. Future Directions

### 11.1. Risk Prediction Models

From identifying high-risk patients who would benefit from screening programs [201,202,203] to estimating SLN status [204,205,206], different melanoma risk prediction models have been proposed in the last decades.

Multiple risk models have been proposed to identify individuals at higher risk for melanoma. They contain established risk factors for melanoma, including age, sex, skin type/color, personal history of skin cancer, number of nevi, presence of dysplastic nevi, and sun exposure history. Models have been proposed for population-level use as self-assessment tools or to be administered by healthcare professionals [202,207,208,209,210,211,212,213,214]. Different studies used varying cutoff values to define high-risk individuals, chosen to optimize sensitivity or specificity. This led to wide variability in the proportion of the population classified as high-risk, with some models flagging up to 55% of the population as high risk when sensitivity was prioritized [215]. Prediction variables from those models are, however, heterogeneous among studies, and study results cannot be easily compared with each other [203,215]. Also, most studies lack external validation. Therefore, international guidelines, including the US Task Force, do not recommend the use of those tools [62]. In clinical practice, a high risk of melanoma is still defined based on individual risk factors.

Different nomograms have also been proposed to predict SLN positivity, such as the Memorial Sloan Kettering Cancer Center [205], the Melanoma Institute of Australia [204], and the Friedman [206] nomograms. Validation studies [216,217,218], however, suggest that these nomograms may underestimate the risk of sentinel lymph node positivity in patients with predicted probabilities ≤10%, limiting their predictive value in lower-risk individuals. As such, caution is advised when using those nomograms to guide clinical decisions.

Risk prediction models are an area of ongoing study, and further validation, as well as improvements in current models, are expected to bring this technology into clinical practice.

### 11.2. Artificial Intelligence-Based Screening Algorithms

In order to improve melanoma diagnostic accuracy and efficiency, as well as to reduce unnecessary skin biopsies, several studies have focused on the development of artificial intelligence (AI)-based skin cancer screening algorithms. Those algorithms evaluate dermoscopic, clinical, or histopathological images. AI-based algorithms can be integrated into various lesion detection methods, using images obtained by clinicians as well as images captured through technologies like total body photography or mobile applications [219].

In the last years, studies have pointed out that AI classifiers can be as sensitive as clinicians in diagnosing melanoma [220,221]. Recently, a meta-analysis of 39 studies [222] evaluated the performance of machine learning, deep learning, and hybrid models (combining Convolutional neural networks, Deep Convolutional neural networks, and neural networks) for melanoma detection. It demonstrated that AI methods excel in classification tasks using dermoscopic images to predict melanoma risk, achieving an area under the curve of 0.96. Hybrid models achieved the highest performances.

Most AI-based algorithm studies, however, have been performed within computational research settings, with few studies applying the technology directly to real patients. AI-based algorithms currently have limited applicability in clinical practice due to the lack of large prospective real-life clinical studies and the reduced number of studies with a diverse dataset. Pre-built databases represent a controlled setting, failing to capture the complexity of real-world scenarios. As evidenced by the EADV Artificial Intelligence (AI) Task Force statement on AI-assisted smartphone apps and web-based services for skin disease [223], the reliability of AI models remains a concern, as they may produce inconsistent results when exposed to diverse data inputs.

Currently, AI can assist clinicians by identifying skin lesions that may require closer monitoring, particularly by detecting early changes. However, AI-based algorithms are not yet a substitute for a clinician’s diagnosis of melanoma or other types of skin cancer. As AI technology and digital health continue to advance rapidly, they hold promising potential to enhance and broaden skin cancer screening strategies in the near future.

### 11.3. Molecular Tests with Prognostic Value

Molecular tests, including gene expression profile [224,225,226,227,228], microRNA expression profile [229], and circulating tumor DNA [230,231], have been proposed to predict the risk of metastasis, disease recurrence, and survival in melanoma patients. MicroRNA and ctDNA are examples of circulating biomarkers analyzed through liquid biopsy, a minimally invasive technique that involves the sampling of tumor-derived material from blood or other biological fluids. Liquid biopsy is gaining increasing importance as a promising tool for the management of cutaneous melanoma.

#### 11.3.1. Gene Expression Profile

Gene expression profile testing measures the messenger RNA expression of a panel of genes extracted from a tumor tissue sample. The first commercially available GEP test for melanoma prognostic purposes was introduced in 2015 [232], evolving into the 31-gene expression profile (31-GEP) test (DecisionDx-Melanoma, Castle Biosciences Inc.). Other current commercially available GEP tests include the MelaGenix 11-GEP test (NeraCare GmbH) and the Merlin Assay CP-GEP test (SkylineDX).

The 31-GEP test has been evaluated in multiple studies for its ability to identify patients at a higher risk of disease recurrence and metastasis [121,233]. It has recently been refined to integrate clinicopathologic features in order to predict SLN positivity [234,235].

Despite promising results, GEP testing still faces limitations, including variability in results among studies with different designs, insufficient long-term patient follow-up, and a lack of studies evaluating GEP test results to guide adjuvant treatment.

Gene expression profiling has also been investigated to improve melanoma diagnostic accuracy. Studies have evaluated its potential to guide biopsy decisions [236] as well as to help distinguish benign melanocytic nevi from malignant melanoma when the histopathologic diagnosis is challenging [237,238].

While GEP testing of melanomas is emerging as a valuable tool for clinical implementation in the foreseeable future, society guidelines still do not recommend its use to guide clinical decision-making for diagnosis, surveillance, SLNB, or adjuvant therapy [107,239]. GEP testing can be considered in the context of clinical trials or in specialized research settings.

#### 11.3.2. microRNA Expression Profile

Multiple studies have investigated the association between circulating microRNA (miRNAs) and melanoma patient survival and disease progression [229,240,241,242,243,244,245]. Studies have also investigated the association between immune-modulatory miRNAs and patient outcomes. A low expression of miR-150, miR-15b, and miR-23a (immune-modulatory miRNAs linked to T cell activation) has been associated with the worst prognosis in melanoma patients [229,245].

Moreover, miRNAs are being investigated for their use in diagnosis [246,247], to predict treatment response and resistance [248,249,250], and as therapeutic targets [251,252,253,254].

A lack of standardization in miRNA analysis, along with difficulties in the implementation of miRNAs as therapeutic targets (such as poor delivery to target tissues, off-target effects, and unwanted toxicity), represents a challenge for the current application of this biomarker in routine clinical practice [253,255,256]. Further and larger studies are needed to incorporate this biomarker into patient care.

#### 11.3.3. Circulating Tumor DNA

Circulating tumor DNA is a promising prognostic biomarker being investigated to predict disease relapse, patient survival, and response to systemic treatment.

ctDNA positivity is often correlated with more advanced disease; therefore, most of the studies on ctDNA in melanoma analyze patients with stages III and IV disease. The detection rate of ctDNA variants in this group of patients can be up to 90% [231,257,258]. In multiple studies, ctDNA positivity, both before treatment and during follow-up, has been shown to be significantly associated with worse recurrence-free and overall survival [231,257,258,259,260,261]. A study by Marsavela et al. [262] evidenced that pretreatment ctDNA predicted progression-free survival only in the first-line immune checkpoint inhibitor treatment setting but not in the second-line immune checkpoint inhibitor setting, especially in patients pretreated with BRAF/MEK inhibitors.

The lack of standardization in ctDNA analysis across different laboratories represents a challenge to the implementation of this biomarker in clinical practice. An additional limitation of plasma ctDNA is its inability to reliably reflect exclusively central nervous system metastases, as the blood–brain barrier restricts the release of ctDNA into the bloodstream [263].

### 11.4. Additional Biomarkers with Prognostic Value

Enzymes such as *N*-methyltransferase (NNMT) and paraoxonase-2 (PON2) are also being investigated for their diagnostic and prognostic value in melanoma. Ganzetti et al. [264] reported higher enzyme nicotinamide *N*-methyltransferase (NNMT) immunoexpression in melanoma cells compared to nevi. This biomarker was further investigated by Sartini et al. [265], who confirmed the upregulation of NNMT in tumor cells and also evidenced higher immunohistochemical expression of NNMT in lymph node metastases compared to both primary melanomas and nevi. The functional role of NNMT was investigated by silencing the enzyme in melanoma cell lines, which, in vitro, was associated with a reduction in melanoma cell proliferation and migration. Enzyme downregulation led to an increase in tumor cell sensitivity to treatment with the chemotherapeutic agent dacarbazine [266]. Another enzyme known to be upregulated in melanoma cells is paraoxonase-2 (PON2) [267]. Its knockdown has similarly been associated, in vitro, with reduced melanoma cell proliferation and viability, as well as increased sensitivity to the chemotherapeutic agent cisplatin [268]. These biomarkers are of significant interest for both prognostic evaluation and therapeutic development. Identifying lesions at high risk of progression remains a critical goal, and, in the future, enzymes such as NNMT and PON2 may also serve as potential targets for molecular-based cancer therapies.

### 11.5. Vaccine as Adjuvant Therapy

Personal neoantigen vaccines are breaking new ground in melanoma treatment. mRNA vaccines encoding tumor-specific neoantigens have been proposed to stimulate the T cell response and immunological memory against melanoma cells [269,270].

Previously, clinical trials for a melanoma vaccine did not show significant improvement in overall and recurrence-free survival rates. Recently, clinical studies have been showing promising results [271,272]. The KEYNOTE-942 phase 2 randomized clinical trial [271] demonstrated reduced risk of recurrence in patients treated with combination mRNA-based individualized neoantigen therapy [mRNA-4157 (V940)] and pembrolizumab versus pembrolizumab monotherapy. Currently, this combination therapy continues to be evaluated in a phase 3 trial (INTerpath-001) as adjuvant treatment of stages IIB–IV resected melanoma [273].

## 12. Conclusions

Cutaneous melanoma is a skin cancer with increased incidence and high mortality. Advances in skin cancer prevention, diagnosis, and treatment are substantially changing disease prognosis. Improvements in melanoma care are expected to continue, driven by innovations in molecular biology and genetics. Understanding current management and the principles behind emerging technologies is crucial to providing excellent patient care. Despite recent developments in diagnosis and management, newer technologies are not always available to the general population, and many patients worldwide do not have access to the latest diagnostic tools and therapies. Alongside pushing the frontier of melanoma care with clinical trials and newer technologies, it is extremely important to continuously educate patients on prevention strategies and highlight the importance of earlier detection of the disease.

## Figures and Tables

**Figure 1 cancers-17-02920-f001:**
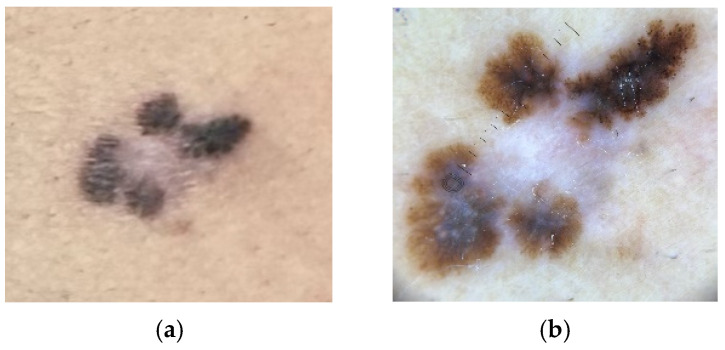
(**a**) Superficial spreading melanoma, gross morphology. (**b**) Dermoscopy of superficial spreading melanoma demonstrating an atypical reticular pattern with radial streaks and a central area of regression.

**Figure 2 cancers-17-02920-f002:**
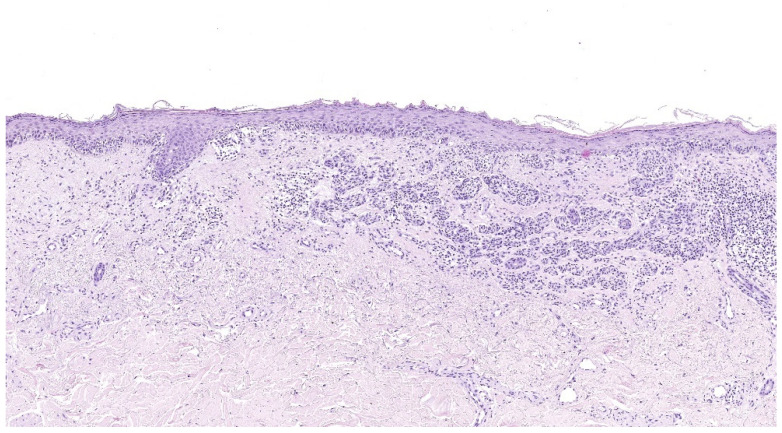
Superficial spreading melanoma, histology (hematoxylin–eosin stain), 10×. Superficial spreading melanoma infiltrating superficial dermis (Breslow thickness 0.7 mm) with moderate non-brisk peritumoral and intratumoral inflammatory infiltrate. (Images courtesy of Dr. Francesca Boggio, Fondazione IRCCS Ca’ Granda Ospedale Maggiore Policlinico, Dermatopathology Laboratory).

**Figure 3 cancers-17-02920-f003:**
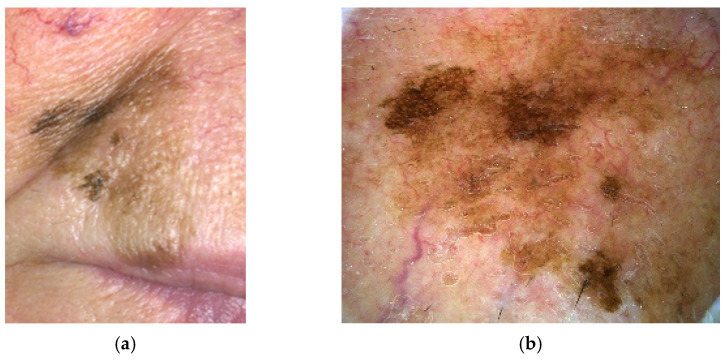
(**a**) Lentigo maligna melanoma, gross morphology. (**b**) Dermoscopy of lentigo maligna melanoma, demonstrating brown and light brown pigmentation with initial obliteration of follicular openings.

**Figure 4 cancers-17-02920-f004:**
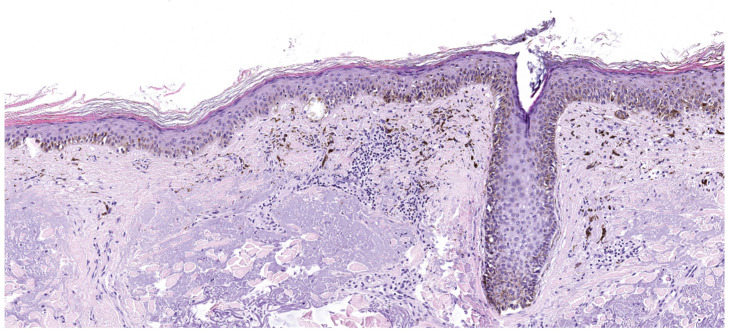
Lentigo maligna, histology, (hematoxylin–eosin stain), 20×. Intraepithelial lentigo maligna with heavily pigmented melanocytes extending to the follicular epithelium and associated with severe solar elastosis in superficial dermis. (Images courtesy of Dr. Francesca Boggio, Fondazione IRCCS Ca’ Granda Ospedale Maggiore Policlinico, Dermatopathology Laboratory).

**Figure 5 cancers-17-02920-f005:**
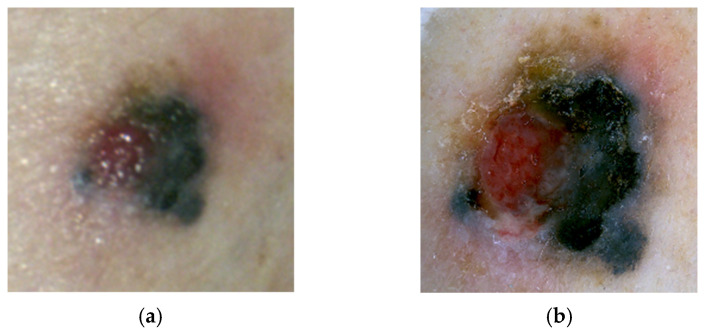
(**a**) Nodular melanoma, gross morphology. (**b**) Dermoscopy of a modular melanoma demonstrating an ulcerated nodule, characterized by blue and black hyperpigmentation. At the periphery, a brown atypical network is visible.

**Figure 6 cancers-17-02920-f006:**
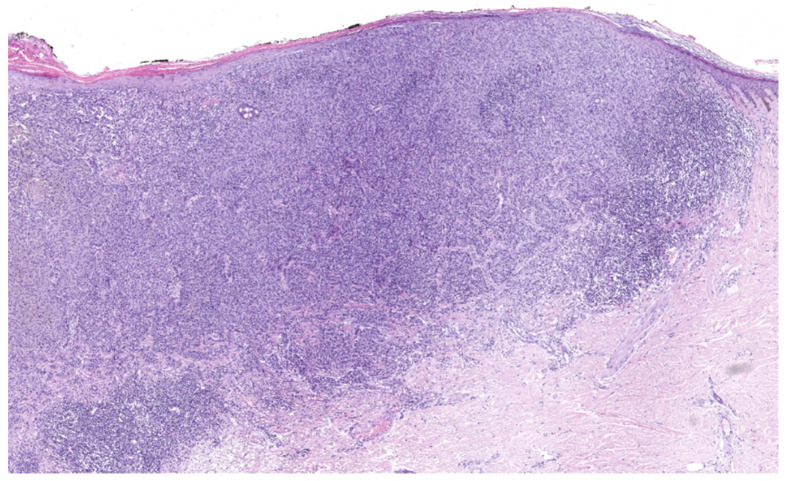
Nodular melanoma, histology (hematoxylin–eosin stain), 10×. Nodular melanoma with densely packed dermal proliferation of atypical melanocytes extending to the reticular dermis (Clark’s level: IV, Breslow thickness 2.9 mm). Several dermal mitoses (10 mitosis × 1 mm^2^) are shown. (Images courtesy of Dr. Francesca Boggio, Fondazione IRCCS Ca’ Granda Ospedale Maggiore Policlinico, Dermatopathology Laboratory).

**Figure 7 cancers-17-02920-f007:**
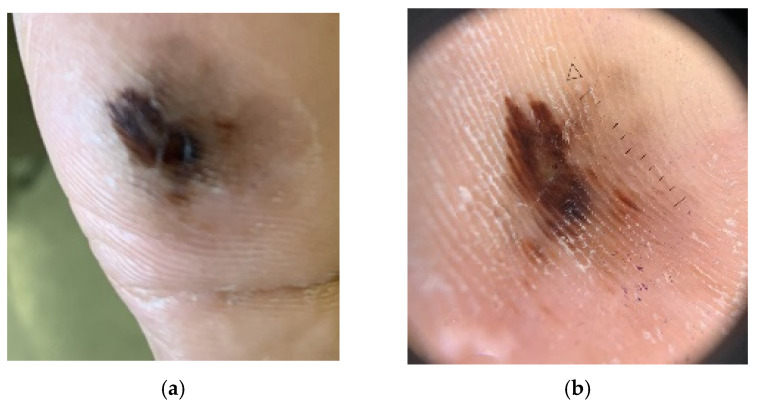
(**a**) Acral lentiginous melanoma, gross morphology. (**b**) Dermoscopy of acral lentiginous melanoma, demonstrating a parallel ridge pattern.

**Figure 8 cancers-17-02920-f008:**
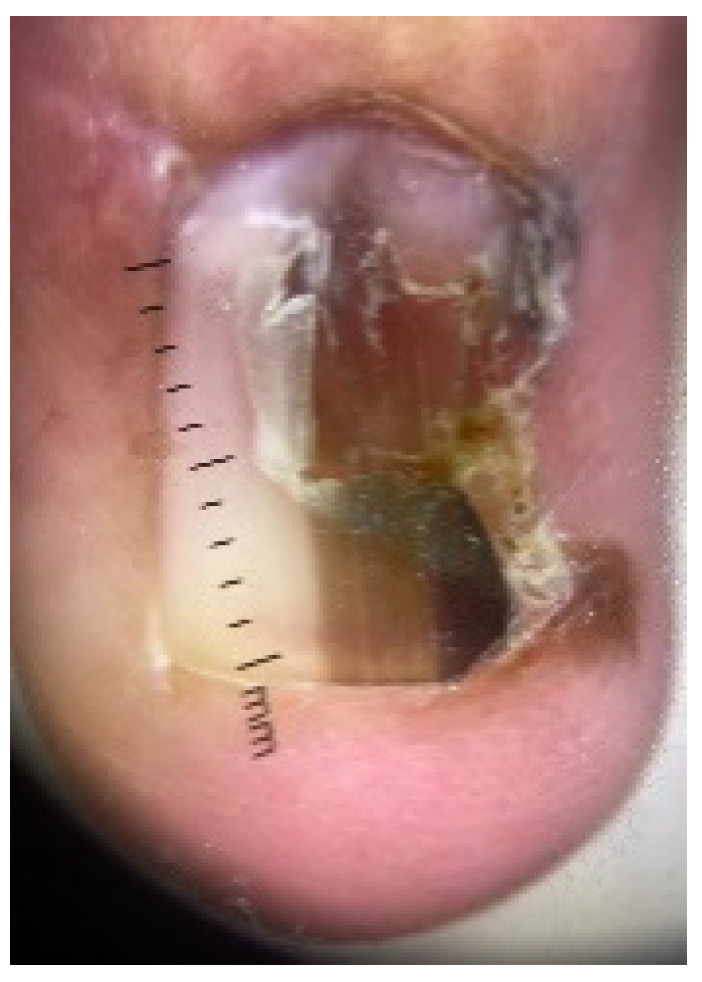
Melanoma in the nail unit, gross morphology. Evidence of nail dystrophy and the involvement of the nail fold (Hutchinson’s sign).

**Figure 9 cancers-17-02920-f009:**
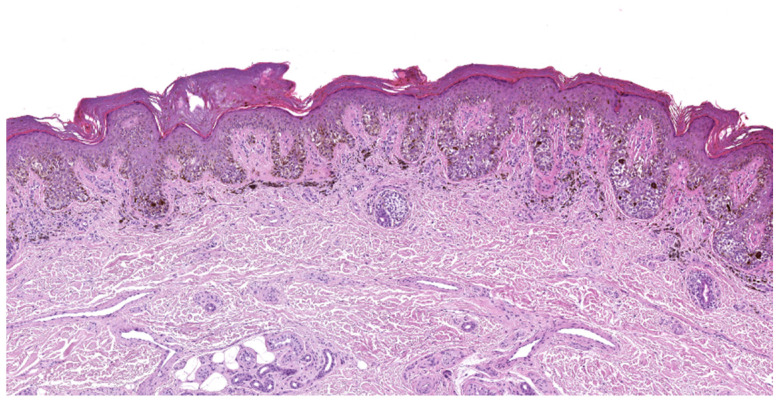
Acral lentiginous melanoma, histology (hematoxylin–eosin stain), 10×. Acral lentiginous in situ melanoma with features of eccrine duct involvement. (Images courtesy of Dr. Francesca Boggio, Fondazione IRCCS Ca’ Granda Ospedale Maggiore Policlinico, Dermatopathology Laboratory).

**Figure 10 cancers-17-02920-f010:**
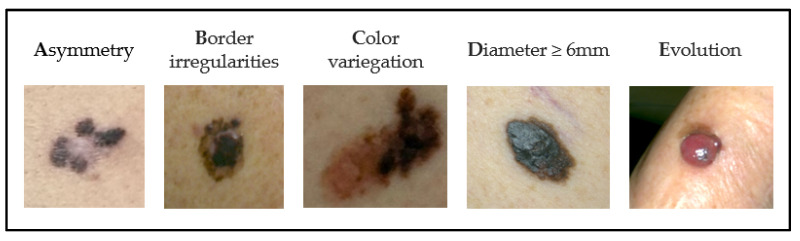
ABCDE criteria for melanoma clinical diagnosis.

**Table 1 cancers-17-02920-t001:** World Health Organization classification of melanocytic tumors based on the multidimensional pathway classification proposed by Bastian et al., introduced in the 4th edition [18].

	Low UV Radiation Exposure/ CSD	High UV Radiation Exposure/ CSD		Low to No (or Variable/Incidental) UV Radiation Exposure/CSD					
**Pathway**	**I**	**II**	**III**	**IV**	**V**	**VI**	**VII**	**VIII**	**IX**
**Pathway Endpoint**	**Low-CSD Melanoma/SSM**	**High-CSD Melanoma/LMM**	**Desmoplastic Melanoma**	**Malignant Spitz Tumor/Spitz Melanoma**	**Acral Melanoma**	**Mucosal Melanoma**	**Melanoma Arising in Congenital Naevus**	**Melanoma in Blue Naevus**	**Uveal Melanoma**
Benign Precursor lesion	Nevus	? IMP	? IMP	Spitz nevus	? Acral nevus	? Melanosis	Congenial nevus	Blue nevus	? Nevus
Intermediate/low-grade dysplasias and melanocytomas	Low-grade dysplasia; BIN; DPN	? IAMP/dysplasia	? IAMP/dysplasia	Atypical Spitz tumor (melanocytoma)	IAMPUS/dysplasia	Atypical melanosis/dysplasia/IAMPUS	Nodule in congenial nevus (melanocytoma)	(Atypical) cellular blue naevus (melanocytoma)	?
Intermediate lesions/high-grade dysplasias and melanocytomas	High-grade dysplasia/MIS; BAP1-inactivated melanocytoma, deep penetrating melanocytoma; PEM; MELTUMP	LM (MIS)	MIS	STUMP/ MELTUMP	Acral MIS	Mucosal MIS	MIS in congenital naevus	Atypical cellular blue nevus	?
Malignant neoplasms	Low-CSD melanoma/SSM (VGP); melanoma in BIN; melanoma in DPN; melanoma in PEM	LMM (VGP)	Desmoplastic melanoma	Malignant Spitz tumor/Spitz melanoma (tumorigenic)	Acral melanoma (VGP)	Mucosal lentiginous melanoma (VGP)	Melanoma in congenital naevus (tumorigenic)	Melanoma in blue naevus (tumorigenic)	Uveal melanoma
Common Mutations *	***BRAF p.V600E*, *MAP2K1*, *NRAS* + *BAP1*/*CTNNb1*/*APC*, *BRAF* + *PRKAR1A*/*PRKCA*** *TERT*, *CDKN2A*, *TP53*, *PTEN*	***NRAS*, *BRAF* (*non-p.V600E*), *KIT***, *NF1* *TERT*, *CDKN2A*, *TP53*, *PTEN*, *RAC1*	***NF1*** *ERBB2*, *MAP2K1*, *MAP3K1*, *BRAF*, *EGFR*, *MET* *TERT*, *NFKBIE*, *NRAS*, *PIK3CA*, *PTPN11*	** *HRAS* ** **, *ALK*, *ROS1*, *RET*, *NTRK1*, *NTRK3*, *BRAF*, *MET*** *CDKN2A*	***KIT*, *NRAS*, *BRAF*, *HRAS*, *KRAS*, *NTRK3*, *ALK*, *NF1***, *CDKN2A*, *TERT*, *CCND1*, *GAB2*	***KIT*, *NRAS*, *KRAS*, *BRAF*** ***NF1*,** *CDKN2*, *SF3B1*, *CCND1*, *CDK4*, *MDM2*	** *NRAS* ** **, *BRAF p.V600E* (small lesions), *BRAF***	***GNAQ*, *GNA11*, *CYSLTR2*** *BAP1*, *EIF1AX*, *SF3B1*	***GNA11*, *GNAQ*, *CYSLTR2*, *PLCB4*** *BAP1*, *SF3B1*, *EIF1AX*

BIN (BAP1-inactivated nevus); CDS (cumulative solar damage); DPN (deep penetrating nevus); IAMP (intraepidermal atypical melanocytic proliferation); IAMPUS (intraepidermal atypical melanocytic proliferation of uncertain significance); IMP (intraepidermal melanocytic proliferation without atypia); LM (lentigo maligna); LMM (lentigo maligna melanoma); MELTUMP (melanocytic tumor of uncertain malignant potential); MIS (melanoma in situ); SSM (superficial spreading melanoma); STUMP (spitzoid tumor of uncertain malignant potential); UV (ultraviolet); VGP (vertical growth phase). * Common mutations in each pathway are listed; mutations already identified in benign or borderline low lesions are shown in bold. Reproduced with permission from WHO Classification of Tumours Editorial Board. Skin tumours [Internet]. Lyon (France): International Agency for Research on Cancer; 2020. (WHO classification of tumours series, 4th ed.; vol. 11 [18]). Available at https://tumourclassification.iarc.who.int/ (accessed on 28 May 2025). David Elder, MBChB.

**Table 2 cancers-17-02920-t002:** Revised seven-point checklist [92].

Clinical Features	Score
Major Criteria	
1. Change in size/new lesion.	2
2. Change in shape/irregular border.	2
3. Change in color/irregular pigmentation.	2
Minor Criteria	
4. Diameter ≥ 7 mm.	1
5. Inflammation.	1
6. Crusting or bleeding.	1
7. Sensory change/itch.	1

A score of 3 or more is considered suspicious for melanoma.

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
