# Peer review of "Melanoma Skin Cancer: A Comprehensive Review of Current Knowledge"

_cancers, 2025, doi:10.3390/cancers17172920_

Round 1

Reviewer 1 Report

Comments and Suggestions for Authors

I have read this review paper on melanoma state of art with great interest, however there are several remarks which should included before possible publication:

  1.    Authors should describe changes in WHO classification of tumours series, 5th ed
  2.    Authors should definitively mention about the limitation of macroscopic systems as ABCDE for diagnosis of early melanoma.
  3.    Authors should also mention is any value of screening for melanoma.
  4.    Line” Prior to lymph node dissection, neoadjuvant therapy is recommended…” I would suggest to change into consider as adjuvant therapy is also recommended option and all patients are candidats for neoadjuvant therapy with immunotherapy.
  5.    Gene names should be written in Italics.
  6.    Authors mentioned Triple immune checkpoint therapy with nivolumab + relatlimab + ipilimumab (RELATIVITY-048 trial), but they did not described aby other investigational fields in metastatic setting as Rp-1 or other – it should be extended.

Author Response

[General Comment] I have read this review paper on melanoma state of art with great interest, however there are several remarks which should included before possible publication:

Response: Thank you for your time in reviewing our paper and for your comments. We really appreciate all the remarks that have been made and we will be modifying our paper accordingly.

[Comment 1] Authors should describe changes in WHO classification of tumours series, 5th ed

Response: We have included the WHO Classification of Skin Tumours 5th edition and explained the similarities and differences from the 4th to 5th edition (pages 7-8, lines 209-228)

[Comment 2] Authors should definitively mention about the limitation of macroscopic systems as ABCDE for diagnosis of early melanoma.

Response: On section 6.1. Clinical Diagnosis (page 11, lines 281-287), we have included information regarding the sensitivity and specificity of the ABCDE criteria and limitations of this system. Regarding limitations, we have pointed out the challenges in diagnosing small lesions (< 6 mm), as well as the recognition of specific melanoma subtypes, such as nodular, amelanotic and desmosplastic melanoma.

[Comment 3] Authors should also mention is any value of screening for melanoma

Response: In the first version of this manuscript, we have initially pointed out in the fifth section (Epidemiology) that routine visual screening exam for melanoma is generally recommended for individuals with high risk for skin cancer. Due to the absence of a randomized melanoma screening trial and no consistent evidence of decrease in mortality due to melanoma populational-based screening programs in observational studies, most healthcare systems do no recommended population-based routine melanoma screening. After reviewing this section, we have decided also to highlight there is no consensus on the ideal melanoma screening strategy and there is insufficient data to recommend melanoma screening for low-risk individuals. We described the importance of screening to improve prognosis and provide a more cost-effective treatment. As well aswe included examples of melanoma screening recommendations in different healthcare settings such as  the United Kingdom, Australia, Germany and Italy (pages 10-11, lines 262-276).

[Comment 4]  Line” Prior to lymph node dissection, neoadjuvant therapy is recommended…” I would suggest to change into consider as adjuvant therapy is also recommended option and all patients are candidats for neoadjuvant therapy with immunotherapy.

Response: Thank you for the suggestion, it definitely clarifies the meaning of the sentence. We have therefore rephrased that sentence to make a clearer statement (page 20, line 549).

[Comment 5]    Gene names should be written in Italics.

Response: Thank you for verifying that, we have now made sure that all gene names are written in Italics. These include: BRAF, NRAS, GNAQ, GNA11, KIT, CDKN2A, XP, TP53, BAP1, NF1, BRCA2, CDK4, MITF, PTEN, POT1 and RB1.

[Comment 6]    Authors mentioned Triple immune checkpoint therapy with nivolumab + relatlimab + ipilimumab (RELATIVITY-048 trial), but they did not described aby other investigational fields in metastatic setting as Rp-1 or other – it should be extended.

Response: Thank you for this suggestion! We have expanded our section on metastatic disease to talk in more details about intralesional/intratumoral therapies (pages 20-21, lines 576-614). We have included detailed information on Talimogene laherparepvec (T-VEC) and vusolimogene oderparepvec (RP1), as well as described other promising oncolytic viral therapies (Canerpaturev C-REV, coxsackievirus A21 - CAVATAK and Lerapolturev – PVSRIPO). We have also briefly explained about cytokine intratumoral injetions, especially Daromun (L19IL2/L19TNF).

Thank you again for your comments and suggestions.

Reviewer 2 Report

Comments and Suggestions for Authors

The manuscript “Melanoma skin cancer: A comprehensive review of current knowledge” is a review article about histopathology, genetics, epidemiology, diagnosis, staging, treatment and implementation of newer technology in the management of cutaneous melanoma.

The article is well-written and easy to read. I really appreciate the work performed by authors. However, some important concerns prevent the publication of the manuscript in the current form:

  1. The abstract could be revised to better underline the state of art and aim of this review.
  2. Clearly explain the novelty and specific contribution of this review compared to existing literature.
  3. The authors should report the limitations of the cited studied and provide a critical evaluation of the reported findings.
  4. Figure 1 and 5 should include scale bar or ruler.
  5. The section of future directions should be expanded taking into account the available literature. For instance, the expression of enzyme nicotinamide N-methyltransferase has been reported as a promising prognostic marker (PMID: 36151433). Moreover, other studies showed that its silencing. Another interesting biomarker could be the enzyme paraoxonase-2. Both have been shown to sensitize melanoma cells to chemotherapeutics.
  6. The section of Artificial Intelligence-based screening algorithms should be expanded better providing the landscape of the current situation and the challenges associated with AI in diagnosis.

Author Response

[General Comment] The article is well-written and easy to read. I really appreciate the work performed by authors. However, some important concerns prevent the publication of the manuscript in the current form. Response: Thank you for your time in reviewing our paper, we have carefully read all the comments and will be making our best to address all of them.

[Comment 1] The abstract could be revised to better underline the state of art and aim of this review. Response: We really appreciated your comment and we have revised the abstract to clarify the current understanding in the field and the objective of the review.

[Comment 2] Clearly explain the novelty and specific contribution of this review compared to existing literature.

Response: With the revisions made to the manuscript, we have aimed to highlight the value of our review to the scientific community. In the rapidly advancing field of cutaneous melanoma, diagnostic and therapeutic innovations are constantly advancing, leading to significant improvements in patient management. However, up-to-date information is often scattered across various sources, making it challenging for clinicians to gain a comprehensive understanding of the current landscape. Our goal is to provide an up-to-date and comprehensive overview of melanoma in a single resource offering a clear understanding of the current state of care, as well as the directions in which emerging technologies are taking melanoma care, and the scientific principles underlying these advancements.

[Comment 3] The authors should report the limitations of the cited studied and provide a critical evaluation of the reported findings.

Response: With the recent revisions added to the manuscript, we have made efforts to include a more critical evaluation of the cited studies.

[Comment 4] Figure 1 and 5 should include scale bar or ruler

Response: We really appreciated your observation, unfortunately we didn’t have available photos that greatly represented SSM and nodular melanomas subtypes together with a scale bar.

[Comment 5] The section of future directions should be expanded taking into account the available literature. For instance, the expression of enzyme nicotinamide N-methyltransferase has been reported as a promising prognostic marker (PMID: 36151433). Moreover, other studies showed that its silencing. Another interesting biomarker could be the enzyme paraoxonase-2. Both have been shown to sensitize melanoma cells to chemotherapeutics.

Response: We really appreciated your observation. In the future directions section, under additional biomarkers with prognostic value (pages 24-25, lines 784-801), we have included additional biomarkers with diagnostic, prognostic, and potential therapeutic relevance. We have included results from the Ganzetti et.al study that reported higher enzyme nicotinamide N-methyltransferase (NNMT) immunoexpression in melanoma cells compared to benign nevi. We have also included evidence from Sartini et. al, showing higher immunohistochemical expression of NNMT in lymph node metastases. Similarly, we added results evidencing paraoxonase-2 upregulation in melanoma cells. We have also included evidence that both NNMT and PON2 may serve as potential targets for molecular-based cancer therapies. As notably pointed out, the downregulation of these enzymes has been linked to increased tumor cells sensitivity to treatment with chemotherapeutic agents. Furthermore, we have detailed the use of other molecular biomarkers such as gene expression profiling, microRNA and ctDNA (pages 23-24, lines 718-783).

[Comment 6] The section of Artificial Intelligence-based screening algorithms should be expanded better providing the landscape of the current situation and the challenges associated with AI in diagnosis

Response: Thank you for your suggestion. We have added to this section an explanation of AI-based models (machine-learning, deep learning and hybrid models) (pages 22-23, lines 689-717). We have also included results from systematic reviews and meta-analysis demonstrating diagnostic accuracy of existing models. In addition to that, we added a paragraph on AI limitations on real-world scenarios and cited concerns reported in the EADV Artificial Intelligence (AI) Task Force statement on AI-assisted smartphone apps and web-based services for skin disease. We have also explained better where AI-models stand in current clinical practice.

Thank you again for your comment and suggestions!

Reviewer 3 Report

Comments and Suggestions for Authors

Dear authors,

It was a pleasure to read such an informative and well-illustrated review. My comments are mostly minor and intended to improve the quality of presentation.

  1. Lines 72–73: Please explain what is meant by “Clark levels III, IV, and V.”
  2. Line 107: Please clarify what degree of sun damage is necessary to induce LMM.
  3. Lines 181–194: Kindly explain why the 4th edition of the WHO Classification of Skin Tumours is discussed, even though the 5th edition has already been released.
  4. Lines 216–217: Please briefly define familial atypical multiple mole and melanoma (FAMMM) syndrome and atypical mole syndrome.
  5. Line 219: Please begin with a lowercase letter and correct the spelling to “Parkinson disease and endometriosis.”
  6. Line 426: Please provide a brief definition of Mohs micrographic surgery.

Author Response

[General Comment] It was a pleasure to read such an informative and well-illustrated review. My comments are mostly minor and intended to improve the quality of presentation.

Response: Thank you for your time in reviewing our paper and for your comments, it really helped us to improve our manuscript!

[Comment 1] Lines 72–73: Please explain what is meant by “Clark levels III, IV, and V.

Response: In the third section (Pathophysiology – page 3, lines 93-104), we have added explanation of Clark levels as well as more information on Breslow staging. Thank you for making this suggestion!

[Comment 2] Line 107: Please clarify what degree of sun damage is necessary to induce LMM.

Response: Thank you for your comment. We have now clarified in section 4.2 (Subtypes: Morphology and Histopathology – page 4, lines 135-136) that Lentigo maligna melanoma arises in sun-damaged skin after cumulative chronic exposure to ultraviolet radiotin.

[Comment 3] Lines 181–194: Kindly explain why the 4th edition of the WHO Classification of Skin Tumours is discussed, even though the 5th edition has already been released.

Response: We have updated to the WHO Classification of Skin Tumours 5th edition and included the similarities and differences from the 4th to 5th edition (pages 7-8, lines 209-228).

[Comment 4] Lines 216–217: Please briefly define familial atypical multiple mole and melanoma (FAMMM) syndrome and atypical mole syndrome.

Response: We have added to fifth section (Epidemiology – page 10, lines 253-258) a brief explanation of the atypical mole syndrome spectrum, as well as specifically explanation on FAMMM syndrome, for better understanding of both terms.

[Comment 5] Line 219: Please begin with a lowercase letter and correct the spelling to “Parkinson disease and endometriosis.”

Response: Thank you for noticing this typographical error. We have corrected it to lowercase letter and correct spelling of Parkinson disease (page 10, line 260).

[Comment 6] Line 426: Please provide a brief definition of Mohs micrographic surgery

Response: Thank you for your suggestion. We have added to the 9.2 section (Management – Surgery – page 18, lines 483-490) an additional paragraph describing briefly Mohs Micrographic Surgery technique and advantages of the procedure.

Thank you again for your comment and suggestions!

Round 2

Reviewer 1 Report

Comments and Suggestions for Authors

Thank you for corrected version, I consider it acceptable in current form

Reviewer 2 Report

Comments and Suggestions for Authors

All raised concerns have been satisfactorily addressed by the authors. The manuscript meets the standards for publication and can be accepted.